# Data linkage of two national databases: Lessons learned from linking the Dutch Arthroplasty Register with the Dutch Foundation for Pharmaceutical Statistics

Heather E. van Brug [1,2]*, Frits R. Rosendaal[2], Liza N. van Steenbergen[3], Rob G. H. H. Nelissen[1,3], Maaike G. J. Gademan[1,2]

1 Department of Orthopaedics, Leiden University Medical Centre, Leiden, The Netherlands, 2 Department of Clinical Epidemiology, Leiden University Medical Centre, Leiden, The Netherlands, 3 Dutch Arthroplasty Register, 's Hertogenbosch, The Netherlands

* h.e.van_brug@lumc.nl

## Abstract

### Background

To provide guidance on data linkage in case of non-unique identifiers, we present a case study linking the Dutch Foundation for Pharmaceutical Statistics and Dutch Arthroplasty Register to investigate opioid prescriptions before/after arthroplasty.

### Methods

Deterministic data linkage was used. Records were linked on: sex, birthyear, postcode, surgery date, or thromboprophylaxis initiation as a proxy for the surgery date. Different postcodes were used, depending on availability: patient postcode (available from 2013 onwards), hospital postcode with codes for physicians/hospitals, and hospital postcode with catchment area. Linkage was assessed in several groups: linked arthroplasties, linked on patient postcode, linked on patient postcode, and low-molecular-weight heparin(LWMH). Linkage quality was assessed by checking prescriptions after death, antibiotics after revision for infection, and presence of multiple prostheses. Representativeness was assessed by comparing the patient-postcode-LMWH group with the remaining arthroplasties. External validation was performed by comparing our opioid prescription rates with those derived from datasets from Statistics Netherlands.

### Results

We linked 317,899 arthroplasties on patient postcode/hospital postcode(48%). Linkage on the hospital postcode appeared insufficient. Linkage uncertainty ranged from roughly 30% in all arthroplasties to 10–21% in the patient-postcode-LMWH-group. This subset resulted in 166.357(42%) linked arthroplasties after 2013 with somewhat younger age, fewer females, and more often osteoarthritis than other indications compared to the other arthroplasties. External validation showed similar increases in opioid prescription rates.

**Data Availability Statement:** Data cannot be shared publicly because of confidentiality. Data are

available from the SFK and LROI Institutional Data Access / Ethics Committee (contact via info@sfk.nl (SFK)/ lroi@orthopeden.org (LROI)/ microdata@cbs.nl (Statistics Netherlands)) for researchers who meet the criteria for access to confidential data. The data underlying the results presented in the study are available from the Foundation of Pharmaceutical Statistics (SFK), the Dutch Arthroplasty Register (LROI) and Statistics Netherlands.

**Funding:** MG received a grant from the Van Rens foundation (Grant number: VRF2019-002). The Van Rens foundation is merged with the Dutch Arthroplasty Register (LROI) in 2020, website funder: https://www.lroi.nl/wetenschap/lroi-subsidie The funders had no role in study design, and analysis or decision to publish. A representative from the LROI is always co-author to oversee the studies in which data of the LROI is used (LvS). LvS fulfilled her duties as co-author and also delivered the data from the LROI. The research group was completely free to make scientific choices and the funder had no role in this. At most LvS put forward suggestions for improvement, which we were free to adopt.

**Competing interests:** The authors have declared that no competing interests exist.

## Conclusions

After identifier selection, checking data availability and internal validity, assessing representativeness, and externally validating our results we found sufficient linkage quality in the patient-postcode-LMWH-group, which consisted of around 42% of the arthroplasties performed after 2013.

## Introduction

Linked data are increasingly used for research purposes, and to enhance and evaluate provided care, [1] for example, by enriching hospital data with mortality registries. Ideally, registries are linked on a unique identifier, ensuring a near-perfect link. This method has frequently been used by the Danish hip arthroplasty register where the possibility exists to link medical databases on individual social security numbers [2]. When linkage on a unique identifier is not possible, due to for instance privacy reasons, non-unique identifiers have to be used [3]. A common approach to linkage on non-unique identifiers is using a combination of variables, such as name, postcode, and date of birth [4], which can also result in a near-perfect link. For instance, Mao et al. demonstrated this in a study where the American Joint Replacement Registry was linked to the administrative data of several hospitals [5]. However, the linkage on non-unique identifiers is often more complicated and prone to errors [4]. Due to the complexity of data linkage with non-unique identifiers, our current study aims to provide guidance on linking data from two sources using non-unique identifiers. Secondly, we describe methods to evaluate the validity of the data linkage, as incorrect links in which two individuals are falsely linked to each other, can also occur. For this purpose, we present a case study in which we linked two national registers. In the methods section, the different steps taken are discussed separately, and each section is introduced with a short explanation of the step. Data linkage was performed as part of a research project in which we aimed to assess opioid prescription rates before and after knee and hip arthroplasty and related outcomes [6].

## Methods

### Data linkage

**Data sources.** Data linkage starts with identifying data sources with sufficient and valid data to answer the research question. We linked the Dutch Arthroplasty Register (LROI) to the Foundation for Pharmaceutical Statistics (SFK). The LROI is a national register including 98% of primary and around 97% of revision arthroplasties [6]. The LROI was our primary data source and consisted of all hip and knee arthroplasties in the Netherlands between 2007–2018. From the LROI variables concerning the arthroplasty surgery were available such as surgery date, survival status (dead/alive), revision surgery (yes/no), the reason for revision, postcode, age, sex, BMI, current smoking status, indication for surgery (osteoarthritis (yes/no), Charnley classification, and ASA classification. The SFK is a national foundation that includes 95% of the community pharmacies, including outpatient pharmacies [7]. The SFK provided prescription data based on the ATC-5 code (analgesics and co-medication). Datasets from Statistics Netherlands (CBS) were used for external validation: opioid reimbursement data [8] and hospital admission data [9].

**Linking.** When linking on multiple identifiers, deterministic or probabilistic linkage methods can be used [10]. Probabilistic linkage uses a probability that determines whether two

**Table 1. Overview of different steps to accomplish the data linkage.**

| | Steps | Description | Example |
|---|---|---|---|
| Preparation | Identifier selection | Selecting variables influenced by discriminatory power, availability of proxy variables, and dynamic variables | Sex, birth year, postcode (dynamic), surgery data, thromboprophylaxis (proxy) |
| | Checking available data and validity | Checking both databases to select additional variables to check the data linkage for implausible situations | Medication after death. Antibiotics after revision. Multiple prostheses linked to different pharmaceutical patients in the linked dataset |
| Internal validation | Checking validity in subgroups | Based on the linkage some subgroups might have better linkage than others, this can be evaluated | Levels of postcode (personal, hospital). Type of thromboprophylaxis (LMWH/DOAC) |
| | Assess representativeness | Compare the linked population to the source population | Check the difference in patient characteristics (age, sex, indication for arthroplasty) |
| External validation | Compare results | Compare the outcomes found in the linked population to the outcomes in a similar setting in an external population | Compare the proportion of arthroplasties with opioid prescriptions in the year of surgery between the linked population and CBS. |

LMWH = Low-molecular-weight heparin, DOAC = Direct oral anticoagulant, CBS = Statistics Netherlands

records belong together. Hence, the researcher can allow for more variability in agreement, for instance, in the case of substantial missing data. This is useful when the data quality is expected to be poor [11]. It, however, involves more complicated algorithms than deterministic linkage. Deterministic linkage is an equally valid method when performed in high-quality data. We expected the data quality to be high and adopted a deterministic linkage method. This compares one or more common identifiers from multiple databases. A link is made when there is agreement on all identifiers [10], as opposed to the probabilistic linkage, in which there is no need for complete agreement. We used sex, birthyear, 4-digit postcode (patient or hospital), operation date, and initiation of thromboprophylaxis. The choice of identifiers should be influenced by discriminatory power, availability of proxy variables, and completeness of identifiers (Table 1). Furthermore, one should be aware of dynamic variables that vary over time. The term discriminatory power refers to the ability of a variable to differentiate between individuals. A proxy variable is a variable that indirectly measures an unavailable variable of interest.

**Proxy variables.** As the SFK holds no surgery dates, a proxy variable had to be identified. Thrombophylactic drugs, prescribed within a two-week time window of the surgery date (4 days before surgery-10 days after surgery), were chosen as all arthroplasty patients routinely receive thromboprophylaxis after knee/hip arthroplasty surgery to prevent thrombotic events [12]. The first and second choices for thromboprophylaxis after arthroplasty are low-molecular-weight heparin (LMWH) and direct-oral anticoagulants (DOACs), both were used for linkage (Table 2).

**Table 2. Information from both registries and their counterpart in the other dataset.**

| SFK | | LROI |
|---|---|---|
| Birth year | | Birth year |
| Sex | | Sex |
| 4-digit postcode (patient) | | 4-digit postcode (patient) |
| Distribution date thromboprophylaxis | | Operation date |
| | LMWH DOAC | |
| 4-digit postcode+ (hospital) | | 4-digit postcode (hospital) |

* combination of apothecary and patient number linked to a unique patient

+ Based on ZIC/AGB-code; LMWH = low-molecular-weight heparin, DOAC = direct-oral anticoagulation

**Choosing identifiers.** Not all identifiers were available for all subjects. Individual patient postcodes were available after 2013. Therefore, for postcode, three different linkage methods were used: 1) linked on patient postcode, 2) linked on hospital postcode combined with unique codes for individual physicians (AGB) or hospitals (ZIC) [13] 3) hospital postcode in combination with the catchment area of the hospital (mean living distance (+1SD) away from hospital (registered since 2014)).

Furthermore, the combination of identifiers should have high discriminatory power. For example, a postcode in the Netherlands consists of 4 letters and 2 numbers. We had the 4-digit postcode to our availability. The median number of residents in a 4-digit postcode area is 2695 (min:5, max:28335) [14]. As anticipated, without including thromboprophylaxis, the combination of the identifiers sex, birthyear, and postcode was not discriminative enough.

Lastly, dynamic variables can have an impact on data linkage. Here, postcode was a dynamic variable, as people may move [15]. Hence, one should ensure that additional information is present for dynamic variables, such as historic postcodes.

In conclusion, we used the following identifiers to link on; sex, birthyear, 4-digit postcode (patient postcode/hospital postcode), and date of surgery combined with thromboprophylaxis.

## Data linkage quality and representativeness

After linking the data, the quality of the linked data should be checked. Incorrect links can be caused by lacking discriminatory power, caused by multiple individuals sharing the same identifiers. Additionally, links might be missed due to data-enter errors, changes in identifiers over time, or when an identifier is missing [16]. Several methods are available to estimate linkage quality. For instance, positive and negative controls can be used to calculate the sensitivity and specificity [16]. But, without knowing the true positive link, these measures are impossible to calculate. However, an indication of the linkage quality and likely linkage errors can be obtained. We propose three steps: 1) estimate the proportion of possible false links by assessing implausible results 2) assess the representativeness against the source population and 3) external validation.

**Estimate the proportion of false links.** We assessed the proportion of possible false links by assessing implausible links in a multi-step procedure. First, we checked if a pseudonymized SFK identifier was linked to more than four primary arthroplasties, as one individual can receive four replacements. Secondly, we assessed the prevalence of medication prescribed after death. We also calculated a corrected percentage of prescriptions after death. Some pharmacists work three weeks ahead, hence medication within the first 21 days after death would not necessarily indicate incorrect linkage. If the number of pills from at least one prescription could be divided by 7, and the prescriptions stopped after that specific prescription we assumed a repeat prescription. Third, the prevalence of antibiotic prescriptions dispensed after revision surgery for infection was assessed as this is common practice [17]. Lastly, we identified the patients who had multiple prostheses in the LROI and checked if these arthroplasties also belonged to the same records in the SFK. The percentage of persons with multiple arthroplasties matched with the SFK was calculated.

We performed these checks in three subgroups. Group 1: all arthroplasties linked on either patient postcode or hospital postcode with ZIC/AGB-code. Group 2: arthroplasties linked on patient postcode. Groups 1 and 2 were identified irrespective of the thromboprophylaxis they were linked on. Group 3: arthroplasties linked on patient postcode and LWMH as thromboprophylaxis. The latter group was identified since other types of thromboprophylaxis, such as DOACs are also given to a large group of patients for chronic anticoagulation treatment.

To check our linkage we considered the patients solely linked on outpatient pharmacy prescriptions as unlinked. The checks described earlier were performed in the subset that was considered linked on a regular pharmacy prescription, except for the check for prosthesis infection, in which we also used arthroplasties linked on outpatient pharmacies.

**Assess the representativeness against the source population.** As the LROI comprises all arthroplasties performed in the Netherlands, we could compare the population that was considered linked to the source population. To assess the possible impact of selecting a subgroup, representativeness was checked. We assessed both patient characteristics (age, sex, BMI, current smoking status) and surgery-related characteristics (indication for surgery (osteoarthritis (yes/no)), Charnley classification, and ASA classification).

**External validation.** We externally validated our study results, the opioid prescription rates, with CBS data. The hospital admission data includes all arthroplasties performed in the Netherlands, except for arthroplasties performed in private hospitals. The medication reimbursement data was used to assess opioid prescriptions (based on Anatomical Therapeutic Chemical code N02A) in the year of arthroplasty surgery. The CBS uses unique identifiers to link the data, as such a near-perfect link is ensured. We compared the proportion of patients with ≥1 opioid prescription in the year of surgery with the proportion found in the linked dataset [18].

**Ethics.** This study was exempted by the Medical Research Ethics Committee Leiden—Den Haag—Delft from review under the Medical Research Involving Human Subjects Act as it did not apply to the current study (number: G19.018). The need for informed consent was waived.

## Results

### Linkage

Arthroplasties linked on hospital postcode in combination with the patient catchment area of the hospital had low discriminatory power and resulted in multiple potential matches per arthroplasty (mean linked pharmaceutical patients per arthroplasty range (1.6–74.2)). We excluded all arthroplasties linked by this technique. Additionally, some arthroplasties were linked to multiple persons in the SFK (17% of arthroplasties linked on patient postcode were linked to >1 person's medication records, and 24% for patients linked on hospital postcode). Using only uniquely linked arthroplasties, we reached an initial linkage percentage of 48.3% of primary and 43.2% of revision arthroplasties operated since 2007 (Fig 1). As indicated above, the arthroplasties solely linked on outpatient pharmacy were considered not useful for our research question on long-term opioid prescriptions (we needed to follow these patients over time), leaving a linkage percentage of 33.5% of primary and 29.4% revision arthroplasties.

**Linkage quality and representativeness.** *Arthroplasties per SFK identifier.* We assessed all arthroplasties that were linked on the patient or hospital postcode. In 294,887 primary linked arthroplasties, 114 SFK identifiers were linked to more than 4 arthroplasties. In group 3 (linked on patient postcode and LMWH) no SFK identifiers were linked to more than 4 arthroplasties. However, 20 of the initially identified 114 SFK identifiers remained present.

*Medication after death.* In group 1, linked on patient postcode or hospital postcode irrespective of prescribed thromboprophylaxis, we found prescribed medication after death in 32.5% of the cases. These percentages decreased to 13.3% in group 3, linked on patient postcode and LMWH (Table 3).

Most medication was dispensed within the first few days after death, indicating registration errors. Furthermore, when correcting for possible repeat prescriptions, the corrected number of medications after death yielded 9.8% after one week, 9.2% after two weeks, and 8.7% after

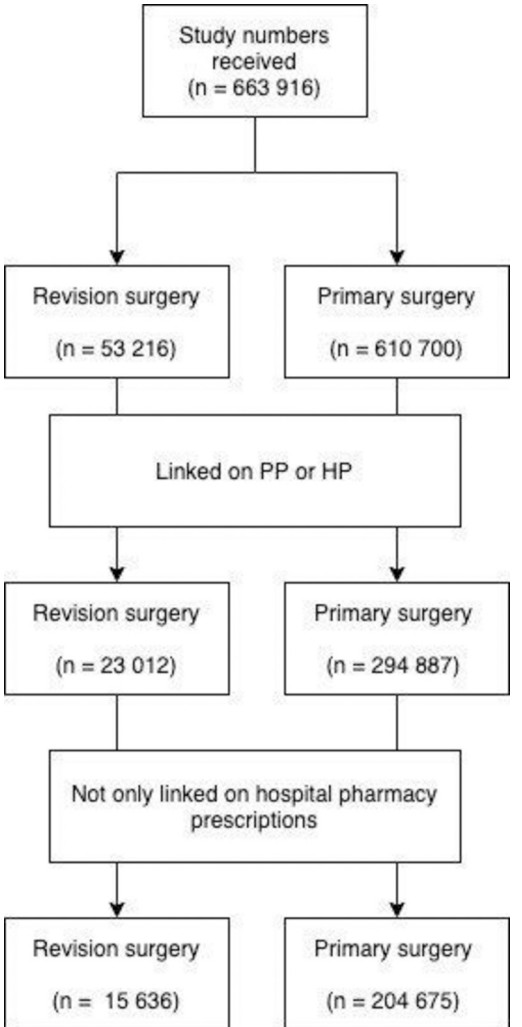

**Fig 1. Flowchart describing the data linkage of primary and revision arthroplasties with data from the SFK.** PP: person postcode; HP: hospital postcode.

**Table 3. Linkage quality checks in different selections of the linked population on several parameters.**

| | Patient and hospital postcode | Patient postcode | Patient postcode and LMWH |
|---|---|---|---|
| **Method** | | | |
| Medication after death | | | |
| Deaths within the study period | 6 851 | 4 918 | 3 270 |
| Medication after death (n(%)) | 2 230 (32.5%) | 1 466 (29.8%) | 435 (13.3%) |
| Antibiotics after revision | | | |
| Partial revision within the study period | 627 | 559 | 326 |
| Antibiotics after (n(%)) | 444 (70.8%) | 402 (71.9%) | 247 (78.2%) |
| Multiple prostheses | | | |
| Multiple prostheses within the study period | 21 384 | 13 223 | 9 552 |
| Match (n*(%)) | 15 192 (71%) | 11 367 (86%) | 8 628 (90.3%) |

n = number of arthroplasties, n* = number of persons

**Table 4. Adjusted and unadjusted percentages for incorrect linkage medication after death.**

| Week | Total | Medication after surgery (n(%)) | Not /7 | Corrected (n(%)) |
|------|-------|--------------------------------|--------|------------------|
| 0 | 3 250 | 435 (13.4%) | - | - |
| 1 | 3 250 | 251 (7.7%) | 69 | 320 (9.8%) |
| 2 | 3 250 | 225 (6.9%) | 6 | 300 (9.2%) |
| 3 | 3 250 | 207 (6.4%) | 2 | 284 (8.7%) |

Total = the number of arthroplasties after which the patient died. Corrected = corrected number of arthroplasties with medication after surgery taking into account the not dividable by 7 prescriptions.

three weeks. We estimated that uncertainty about linkage remained present for approximately 10% of the linked arthroplasties (Table 4).

*Antibiotics after prosthetic joint infection.* In group 1, we found that 70.8% of patients with a partial revision for infection received antibiotics after surgery, which increased up to 78.2% when linked on patient postcode and LMWH (Table 3).

*Multiple prostheses.* In the overall LROI population, 71% of the patients with multiple prostheses were also identified as such in the SFK dataset which increased to 91.3% in the patient postcode LMWH group (group 3) (Table 3).

*Assessment linkage checks.* The proportion of possible false links varied between the different linkage groups. The group linked on patient postcode, therefore arthroplasties performed after 2013, and LMWH had the best-estimated data quality. This group consisted of 153.808 (41.7%) linked primary arthroplasties and 12.549 (39.6%) linked revision arthroplasties performed since 2013. This subset included 103.943 primary and 7.848 revision arthroplasties also linked to a community pharmacy, which we were able to use in our research into long-term opioid prescriptions. In this group, we estimate a properly linked match in approximately 90% of the arthroplasties.

*Representativeness of the linked data.* We found that the characteristics of the group linked on patient postcode and LMWH (group 3), compared to the characteristics of the other arthroplasties (non-linked and other linked arthroplasties) differed slightly. Our linked population included more knee arthroplasties, was slightly younger, less often female, more often had osteoarthritis as an indication, and had a somewhat lower Charnley Score and ASA classification than unlinked arthroplasties (supplementary table 1A-B). Similar results were seen when we compared the population selected for our study on long-term opioid prescriptions (excluding the arthroplasties linked on outpatient pharmacy), to the other, less well-linked, population [18].

*External validation.* In the database of the CBS, the prevalence of opioid prescriptions in the year of operation slightly differed from the prevalence found in our group 3. The difference in percentage points ranged from 1.8% in 2016 to 9.1% in 2018. In both datasets, the proportion of patients with at least one opioid prescription increased at similar rates between 2013 and 2018 [18].

## Discussion and concluding remarks

To provide guidance on data linkage we presented a case study in which we linked the SFK and the LROI to investigate opioid prescriptions before/after arthroplasty without a unique identifier. We linked the two datasets based on deterministic linkage and found an initial linkage rate of 43–48%. We presented the data linkage quality using several methods and were able to evaluate representativeness and externally validated our results.

We found that the linkage uncertainty percentage ranged between 10–22%. Linkage uncertainty was highest when we assessed the proportion of revisions for an infection that did not

receive any antibiotic. Besides mismatch, misclassification could be an issue here. Recently it was shown that 10% of arthroplasties recorded as having a revision due to an infection in the LROI, did not show up in an infection database [19]. Secondly, due to the lack of an overarching unique identifier for pharmacies before 2017, antibiotic prescriptions might not be assigned to a specific patient if these were collected in the outpatient pharmacy [7]. Hence, the 22% linkage uncertainty will likely be an overestimation. Overall, we concluded that a linked dataset could be generated with an estimated 90% linkage certainty, which we deemed acceptable.

Several recent studies have tried to provide guidance on the reporting and evaluation of data linkage [4, 20–22]. However, quality assessment for the linkage certainty was not mentioned, as no cut-off values exist and the quality depends on the used identifiers and their discriminatory power. For instance, in other research, using surname, date of birth, and sex over 95% linkage certainty was accomplished [23]. Ways to evaluate the impact of the uncertainty, such as comparison of linked and unlinked cases or comparisons with external databases should be explored [4].

For our main research project on opioid prescription rates before/after knee and hip arthroplasty, we selected the population in which we were most certain of our data linkage. However restricting the linked population can result in a specific subgroup, which may not be representative of the source population [24]. As such, it might be reasonable to accept less linkage certainty, when the representativeness of data after linkage is paramount.

Additionally, external validation of the study results assures reproducibility and generalizability [25]. However, this might pose an issue as an external source should be available and representative of your sample [16]. Moreover, the dataset available might contain different data.

A limitation in our data linkage is that, due to legislation, we could not assess which identifiers were missing from the sample which created difficulty to link. However, we know that patients who did not collect their thromboprophylaxis within the dedicated timeframe were not eligible for linkage. This concerns patients with prolonged hospital stays or patients discharged to nursing homes. Additionally, arthroplasties linked to multiple persons were excluded. These could have represented elderly living in similar areas with the same 4-digit postcode. Furthermore, because the patient postcode, for most arthroplasties, was available after 2013, we were unable to link the arthroplasties performed before 2013.

In conclusion, partial identifiers and well-defined proxies as patient identifiers, in combination with additional variables, can be used to uniquely link different databases. However, sufficient data should be available to check the data linkage quality. Assessing the representativeness of the study population is a must to evaluate the effect of selecting better-linked subgroups. Using external validation is necessary for the evaluation of the generalizability and reproducibility of the study results. By following the steps described in the current study we were able to conclude sufficient linkage quality in our linked population with 42% of the arthroplasties, performed after 2013, linked. We advise others, to ensure high-quality research or patient care, in the case of linked data, to carefully consider each of the mentioned steps before starting the data linkage.

## Supporting information

**S1 Table. The selected primary arthroplasty population linked on low-molecular-weight heparin and patient postcode compared to the not-linked population.**
(DOCX)

**S2 Table. The selected revision arthroplasty population linked on low-molecular weight heparin and patient postcode compared to the not-linked population.**
(DOCX)

## Author Contributions

**Conceptualization:** Frits R. Rosendaal, Liza N. van Steenbergen, Rob G. H. H. Nelissen, Maaike G. J. Gademan.

**Data curation:** Liza N. van Steenbergen.

**Formal analysis:** Heather E. van Brug.

**Funding acquisition:** Rob G. H. H. Nelissen, Maaike G. J. Gademan.

**Methodology:** Heather E. van Brug, Frits R. Rosendaal, Rob G. H. H. Nelissen, Maaike G. J. Gademan.

**Project administration:** Heather E. van Brug, Maaike G. J. Gademan.

**Software:** Heather E. van Brug.

**Supervision:** Rob G. H. H. Nelissen, Maaike G. J. Gademan.

**Validation:** Heather E. van Brug, Maaike G. J. Gademan.

**Visualization:** Heather E. van Brug.

**Writing – original draft:** Heather E. van Brug, Maaike G. J. Gademan.

**Writing – review & editing:** Heather E. van Brug, Frits R. Rosendaal, Liza N. van Steenbergen, Rob G. H. H. Nelissen, Maaike G. J. Gademan.

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
