## [Decision Letter · Decision Letter 0]

24 Aug 2022

PONE-D-22-07971Data linkage of two national databases: lessons learned from linking the Dutch Arthroplasty Register with the Dutch Foundation for Pharmaceutical StatisticsPLOS ONE

Dear Dr. van Brug,

Thank you for submitting your manuscript to PLOS ONE. After careful consideration by an external reviewer and an Academic Editor, we feel that it has merit but does not fully meet PLOS ONE’s publication criteria as it currently stands. Therefore, we invite you to submit a revised version of the manuscript that addresses the points raised during the review process.

We look forward to receiving your revised manuscript.

Kind regards,

Shao-Hsien Liu

Academic Editor

PLOS ONE

https://journals.plos.org/plosone/s/file?id=ba62/PLOSOne_formatting_sample_title_authors_affiliations.pdf".

2. In the ethics statement in the Methods and online submission information, please ensure that you have specified what type you obtained (for instance, written or verbal, and if verbal, how it was documented and witnessed). If your study included minors, state whether you obtained consent from parents or guardians. If the need for consent was waived by the ethics committee, please include this information.

Editor Comments:

Dear Authors,

I would like to thank you for submitting the manuscript “Data linkage of two national databases: lessons learned from linking the Dutch Arthroplasty Register with the Dutch Foundation for Pharmaceutical Statistics” to PLUS ONE. This manuscript presented an example of how researchers can utilize and link different databases for clinically relevant research in patients with total joint arthroplasty.

This is a clinically relevant study since the lessons learned from this data linkage process are important for future applications. The strengths of the study include the use of large database and population data and exploration of several data linkage approaches. However, I still have some questions about a couple places and approaches.

It is not clear what the external validation meant here in the present manuscript. In general, external validation means to evaluate the extent to which the findings of a study can be applicable to other settings. Despite that the current metrics used to compared findings for the main purpose/study questions of the data merging were relevant, the representativeness and/or characteristics of the merged sample compared to other studies were not provided. In addition, generalizability requires internal validity as well as a judgement on whether the findings of a study are applicable to a particular group. However, such discussion in the current manuscript seems lacking.

I also recommend having this manuscript reviewed with editing process since some places should be written in past tense. Please consider having someone review the grammar on this manuscript.

Please see below for the specific comments.

 

Abstract:

• Line 27, “To provide guidance on data linkage in case of non-unique identifiers we present….” – suggest adding a coma before we

• Line 38-39, “Representativeness was assessed by comparing the patient postcode-LMWH-group with the other arthroplasties” – what are other arthroplasties that were referring to? Please be specific.

• Line 39-40, it is not clear what the external validation meant? Was it referring to the representativeness of the overall sample from the data linkage compared to other sample (generalizability) or any other outputs that authors compared with?

• Line 41, we typically do not use numbers to start with a sentence.

• Instead of listing your findings, conclusion should present your main results in light of the evidence in your research. Please consider revising the conclusion to reflect main findings as well as the specific strengths and limitations of your research.

Introduction:

• It would benefit the readers if authors can provide additional context on what are the common approaches for non-unique identifiers to link databases and how they typically perform. So that later when author present your findings, readers can follow through with better ideas about those approaches.

• It would be easier for readers to follow if authors can separate the long paragraphs into different sections.

Methods:

• It is not clear what the external validation meant here in the present manuscript. In general, external validation means to evaluate the extent to which the findings of a study can be applicable to other settings. Despite that the current metrics used to compared findings for the main purpose/study questions of the data merging were relevant, the representees and/or characteristics of the merged sample compared to other studies were not provided. In addition, generalizability requires internal validity as well as a judgement on whether the findings of a study are applicable to a particular group. However, such discussion in the current manuscript seems lacking. A section to describe the process of evaluating internal validation may be helpful.

• I would also suggest a separate section for the description of validation process that includes the discussion of potential metrics and/or sample characteristics to evaluate the representativeness of sample from the merged database.

• The rationale behind probabilistic versus deterministic linkage was not clear. For example, probabilistic linkage could be a more accurate method in poorer quality data. However, such context or information was not provided in the paper.

• The information regarding the specific discriminative power (from basic set to additional variables added) were not available in the paper. To our knowledge, the performance of deterministic linkage could be impacted by reduced discriminative power. Information on how much the discriminative power were improved could be beneficial to the readers.

• The rationale behind the of use thromboprophylaxis as a potential identifier was not clear.

• Line 160-168, this section reads more like background information on why authors used those measures to assess quality of data linkage rather than the description of approaches. Those may be more appropriate in the intro or discussion sections.

Results:

• Based on the findings from initial linkage rate which was about 43-48%, I wonder authors have information on what main identifiers were missing from the sample that those data could not be linked?

• Line 289-291, the results corresponding to the description here were not provided anywhere in the manuscript.

• Line 293-298, same comments above. Either a figure and/or numbers representing that information was missing.

Discussion:

• The first paragraph of discussion (line 308-326) mainly focused on the 10% linkage uncertainty from group 3. However, it would be helpful that authors add appropriate references/citations to support your explanations or arguments here in the section.

• Line 336-351, is there any data or published information to support the findings from CBS data regarding opioid prescriptions. Currently, no data was presented or provided with a reference to support those discussions.

Tables & Figures:

• Some components belong to the evaluation of internal validation, and some are for external validation. It would be helpful to provide subheading with corresponding sections on those steps regarding the data linkage process.

• Table 3: instead of group 1, 2 ,3, it would be beneficial to the readers if authors can list/name those groups directly in the column.

I also recommend having this manuscript reviewed with editing process since some places should be written in past tense. Please consider having someone review the grammar on this manuscript.

Reviewers' comments:

Reviewer's Responses to Questions

**Comments to the Author**

1. Is the manuscript technically sound, and do the data support the conclusions?

Reviewer #1: Yes

2. Has the statistical analysis been performed appropriately and rigorously? 

Reviewer #1: Yes

3. Have the authors made all data underlying the findings in their manuscript fully available?

Reviewer #1: No

4. Is the manuscript presented in an intelligible fashion and written in standard English?

Reviewer #1: Yes

5. Review Comments to the Author

Reviewer #1: This manuscript is very well-written and addresses an important topic in registry science. The biggest concern is that the analysis assumes that prescription medications are used for VTE prophylaxis in arthroplasty cases. The authors write, “As the SFK holds no surgery dates, a proxy variable had to be identified. Thromboprophylactic drugs prescribed within a two week time window of the surgery date (4 days before surgery – 10 days after surgery) was chosen as arthroplasty patients routinely receive thromboprophylaxis after knee/hip arthroplasty surgery to prevent thrombotic events.” [lines 124-128] In the United States, for example, there is empirical evidence that there has been a dramatic increase in the use of aspirin for VTE prophylaxis since 2013 for knee replacements (Hood et al., JAMA, 2019) and hip replacements (Muscatelli et al., Journal of Arthoplasty, 2021). Thus, the authors would strengthen the analysis by providing evidence that prescription chemoprophylaxis for VTE is indeed routine for virtually all patient in the Netherlands for arthroplasty.

6. PLOS authors have the option to publish the peer review history of their article (what does this mean?). If published, this will include your full peer review and any attached files.

Reviewer #1: No

---

## [Author Response · Author response to Decision Letter 0]

7 Oct 2022

Dear reviewer, 

We've answered all raised questions in the attached "response to reviewers" documents. We hope this is sufficient. 

Kind regards, 

Heather van Brug

---

## [Decision Letter · Decision Letter 1]

26 Dec 2022

PONE-D-22-07971R1Data linkage of two national databases: Lessons learned from linking the Dutch Arthroplasty Register with the Dutch Foundation for Pharmaceutical StatisticsPLOS ONE

Dear Dr. van Brug,

Thank you for submitting your manuscript to PLOS ONE. After careful consideration, we feel that it has merit but does not fully meet PLOS ONE’s publication criteria as it currently stands. Therefore, we invite you to submit a revised version of the manuscript that addresses the points raised during the review process.

We look forward to receiving your revised manuscript.

Kind regards,

Dimitrios Sokratis Komaris, Ph.D

Academic Editor

PLOS ONE

Reviewers' comments:

Reviewer's Responses to Questions

**Comments to the Author**

1. If the authors have adequately addressed your comments raised in a previous round of review and you feel that this manuscript is now acceptable for publication, you may indicate that here to bypass the “Comments to the Author” section, enter your conflict of interest statement in the “Confidential to Editor” section, and submit your "Accept" recommendation.

Reviewer #1: (No Response)

Reviewer #2: (No Response)

2. Is the manuscript technically sound, and do the data support the conclusions?

Reviewer #1: Yes

Reviewer #2: Partly

3. Has the statistical analysis been performed appropriately and rigorously? 

Reviewer #1: Yes

Reviewer #2: N/A

4. Have the authors made all data underlying the findings in their manuscript fully available?

Reviewer #1: No

Reviewer #2: No

5. Is the manuscript presented in an intelligible fashion and written in standard English?

Reviewer #1: Yes

Reviewer #2: No

6. Review Comments to the Author

Reviewer #1: In my initial review I expressed concern about the assumption that all TKA and THA patients get chemoprophylaxis for VTE. I indicated that this claim should be supported because elsewhere in the world there has been a large shift to aspirin, which would not show up in SFK data. The authors seem to have responded by adding Reference 13 on line 133. However, reference 13 appears to be "An analysis of the relocation behavior of people in their sixties" that is not relevant to the VTE prophylaxis issue. The reference should be corrected.

Reviewer #2: Dear Editor, Dear Author

First the subject of the article is of great interest for the future use of Big data in healthcare surveillance and research.

However, the writing and understanding of the research performed could be synthetized and hence clarified. Indeed, the second sentence has no verb! In the second paragraph the tense of the verbs must also be homogenized.

Moreover, Netherlands is lucky with his unique patient number over the country for most of their healthcare databases. Which is not the case in a lot of European countries.

And the key issue of non-unique identifiers is less important in their reuse as compared with other countries.

The definition of case study, in informatics we would probably call it use case in orthopedics more than case study, in my opinion.

The article is interesting but must been shortened and synthetize to go straight to the point, with less chapter titles and short paragraphs.

The discussion must also be shortened, to be suitable to publish as a use case.

7. PLOS authors have the option to publish the peer review history of their article (what does this mean?). If published, this will include your full peer review and any attached files.

Reviewer #1: No

Reviewer #2: **Yes: **Leslie Grammatico-Guillon

---

## [Author Response · Author response to Decision Letter 1]

6 Jan 2023

We have uploaded a file that shows all responses.

---

## [Decision Letter · Decision Letter 2]

17 Feb 2023

Data linkage of two national databases: Lessons learned from linking the Dutch Arthroplasty Register with the Dutch Foundation for Pharmaceutical Statistics

PONE-D-22-07971R2

Dear Dr. van Brug,

We’re pleased to inform you that your manuscript has been judged scientifically suitable for publication and will be formally accepted for publication once it meets all outstanding technical requirements.

Kind regards,

Dimitrios Sokratis Komaris, Ph.D

Academic Editor

PLOS ONE

Additional Editor Comments (optional):

Reviewers' comments:

Reviewer's Responses to Questions

**Comments to the Author**

1. If the authors have adequately addressed your comments raised in a previous round of review and you feel that this manuscript is now acceptable for publication, you may indicate that here to bypass the “Comments to the Author” section, enter your conflict of interest statement in the “Confidential to Editor” section, and submit your "Accept" recommendation.

Reviewer #1: All comments have been addressed

2. Is the manuscript technically sound, and do the data support the conclusions?

Reviewer #1: Yes

3. Has the statistical analysis been performed appropriately and rigorously? 

Reviewer #1: Yes

4. Have the authors made all data underlying the findings in their manuscript fully available?

Reviewer #1: No

5. Is the manuscript presented in an intelligible fashion and written in standard English?

Reviewer #1: Yes

6. Review Comments to the Author

Reviewer #1: (No Response)

7. PLOS authors have the option to publish the peer review history of their article (what does this mean?). If published, this will include your full peer review and any attached files.

Reviewer #1: No

---

## [Editor Report · Acceptance letter]

27 Feb 2023

PONE-D-22-07971R2 

Data linkage of two national databases: Lessons learned from linking the Dutch Arthroplasty Register with the Dutch Foundation for Pharmaceutical Statistics 

Dear Dr. van Brug:

I'm pleased to inform you that your manuscript has been deemed suitable for publication in PLOS ONE. Congratulations! Your manuscript is now with our production department. 

Kind regards, 

on behalf of

Dr. Dimitrios Sokratis Komaris 

Academic Editor

PLOS ONE